# Electrically induced cancellation and inversion of piezoelectricity in ferroelectric Hf$_{0.5}$Zr$_{0.5}$O$_2$

Haidong Lu [1,7], Dong-Jik Kim[2,7], Hugo Aramberri [3], Marco Holzer [2,4], Pratyush Buragohain [1], Sangita Dutta [3,5], Uwe Schroeder [6], Veeresh Deshpande[2], Jorge Íñiguez [3,5] ✉, Alexei Gruverman [1] ✉ & Catherine Dubourdieu [2,4] ✉

HfO$_2$-based thin films hold huge promise for integrated devices as they show full compatibility with semiconductor technologies and robust ferroelectric properties at nanometer scale. While their polarization switching behavior has been widely investigated, their electromechanical response received much less attention so far. Here, we demonstrate that piezoelectricity in Hf$_{0.5}$Zr$_{0.5}$O$_2$ ferroelectric capacitors is not an invariable property but, in fact, can be intrinsically changed by electrical field cycling. Hf$_{0.5}$Zr$_{0.5}$O$_2$ capacitors subjected to ac cycling undergo a continuous transition from a positive effective piezoelectric coefficient $d_{33}$ in the pristine state to a fully inverted negative $d_{33}$ state, while, in parallel, the polarization monotonically increases. Not only can the sign of $d_{33}$ be uniformly inverted in the whole capacitor volume, but also, with proper ac training, the net effective piezoresponse can be nullified while the polarization is kept fully switchable. Moreover, the local piezoresponse force microscopy signal also gradually goes through the zero value upon ac cycling. Density functional theory calculations suggest that the observed behavior is a result of a structural transformation from a weakly-developed polar orthorhombic phase towards a well-developed polar orthorhombic phase. The calculations also suggest the possible occurrence of a non-piezoelectric ferroelectric Hf$_{0.5}$Zr$_{0.5}$O$_2$. Our experimental findings create an unprecedented potential for tuning the electromechanical functionality of ferroelectric HfO$_2$-based devices.

The discovery of ferroelectricity in hafnium oxide (HfO$_2$) thin films opened exciting opportunities for their monolithic integration with the existing complementary metal-oxide-semiconductor (CMOS) technology[1,2]. Ever since the first report of ferroelectricity in Si-doped HfO$_2$ (Si:HfO$_2$)[1], there have been concerted efforts towards understanding the origin of ferroelectricity in hafnium oxide[3–10]. Ferroelectricity has been primarily attributed to a polar orthorhombic phase (space group $Pca2_1$, noted o-III phase)[4,5,11], although a polar

[1]Department of Physics and Astronomy, University of Nebraska-Lincoln, Lincoln, NE 68588-0299, USA. [2]Helmholtz-Zentrum Berlin für Materialien und Energie, Insitute Functional Oxides for Energy-Efficient Information Technology, Hahn Meitner Platz 1, 14109 Berlin, Germany. [3]Materials Research and Technology Department, Luxembourg Institute of Science and Technology (LIST), Avenue des Hauts-Fourneaux 5, L-4362 Esch/Alzette, Luxembourg. [4]Freie Universität Berlin, Physical and Theoretical Chemistry, Arnimallee 22, 14195 Berlin, Germany. [5]Department of Physics and Materials Science, University of Luxembourg, Rue du Brill 41, L-4422 Belvaux, Luxembourg. [6]NaMLab gGmbH, Noethnitzer Strasse 64 a, 01187 Dresden, Germany. [7]These authors contributed equally: Haidong Lu and Dong-Jik Kim. ✉e-mail: jorge.iniguez@list.lu; agruverman2@unl.edu; catherine.dubourdieu@helmholtz-berlin.de

rhombohedral phase (*R3m*, r-phase) has also been experimentally reported[7] and at least five polar phases have been predicted by density functional theory (DFT)[12]. Due to the relatively small energy differences between the various polymorphs of $HfO_2$, there can be phase coexistence of centrosymmetric monoclinic (*P2₁/c*, m-phase) and tetragonal (*P4₂/mnc*, t-phase) phases along with the polar non-centrosymmetric phases in the polycrystalline $HfO_2$ films, leading to a complex system that is highly sensitive to the growth conditions, doping concentration, strain, film thickness, electrodes, etc.[3,10,13–17]. Consequently, complex behaviors such as antiferroelectricity in t-phase rich films[2,3], field-induced phase transformations and enhanced ferroelectricity with field cycling[11,18,19], interplay between depolarization field and wake-up effects[20], are frequently observed in $HfO_2$-based ferroelectric (FE) systems. In addition, several intriguing features, such as the disputed nature of ferroelectricity[9,21–23], enhancement of ferroelectricity at reduced dimensions[2], negative domain wall energies[9,24] and negative longitudinal piezoelectric coefficient, $d_{33}$[25–28], make hafnia ferroelectrics a unique and fascinating class of materials to study.

The potential of FE $HfO_2$ towards technological applications has been demonstrated already through the realization of ferroelectric field effect transistors (FeFETs)[29], 3-dimensional trench capacitors with high endurance[30], negative capacitance-based devices that allow transistors to overcome "Boltzmann's tyranny"[31], ferroelectric tunnel junctions (FTJs) for resistive switching[32–34] and neuromorphic computing applications[35–37]. The recent surge of interest in the piezoelectric properties of these materials stems from the emerging controversy between the theoretical and experimental data. Several theoretical reports have predicted a negative longitudinal piezocoefficient $d_{33}$[25–27], which arises from the peculiar chemical environment of the oxygen atoms. However, the majority of the experimental works have demonstrated a positive $d_{33}$[38–41]. Only recently, a few experimental demonstrations of negative $d_{33}$ in $HfO_2$ have been reported[27,28,42]. Moreover, it has been observed that the sign of the $d_{33}$ could vary depending on the film thickness, electrode materials, and deposition method used[38–42]. In addition, the coexistence of regions with positive and negative $d_{33}$ piezocoefficients within the same device has been reported[42].

From the experimental point of view, the sign of the piezoelectric coefficient determines whether the ferroelectric sample expands or contracts under an applied electric field—a phenomenon known as the converse piezoeffect. One of the most common methods to investigate this phenomenon is piezoresponse force microscopy (PFM). This technique employs a conductive nanoscopic tip under an electrical bias to probe the local electromechanical response of the sample at the scale range of several nanometers, thereby providing information on the local piezoelectric properties[43]. Application of PFM has revealed the existence of regions in the $Hf_{0.5}Zr_{0.5}O_2$ (HZO) films that were apparently switching against the external electric field, as was deduced from monitoring the change in the PFM phase signal[42,44]. However, careful analysis of the calibrated piezoresponse signal demonstrated that this apparently abnormal switching behavior was, in fact, a manifestation of the regions with positive $d_{33}$ coefficient within the negative $d_{33}$ matrix[42].

Here, we show that the sign of piezoelectricity in the HZO films can be changed by an electrical field and, in addition, that piezoelectricity can be nullified while keeping robust switchable polarization. PFM imaging of the W/HZO/W capacitors subjected to ac cycling revealed a gradual transition from a uniformly positive $d_{33}$ state in the pristine capacitors to a fully inverted uniform negative $d_{33}$ state (while, in parallel, the polarization monotonically increases by about 20%). At some intermediate stage along this evolution, the HZO capacitors exhibit a zero net piezoelectricity due to coexisting positive and negative $d_{33}$ regions while displaying fully switchable polarization. Local PFM spectroscopy provides an additional insight into this process by displaying a gradual reduction of the positive $d_{33}$ value, and then a gradual increase in the $d_{33}$ value but with a negative sign. Remarkably, every single location of the capacitor goes through zero local piezoelectricity upon ac cycling. DFT calculations investigating the path from the non-polar tetragonal phase to the ferroelectric o-III phase reveal an unprecedented non-monotonic behavior of the strain with the development of polarization. Hence, the simulations suggest that the initial positive $d_{33}$ sign and sizeable remanent polarization ($\geq$ 19 $\mu C/cm^2$) in the pristine state can be attributed to a weakly-developed polar orthorhombic phase that, upon ac cycling, progressively evolves towards a well-developed polar o-III phase with negative $d_{33}$.

## Results

### Ferroelectricity switching in W/HZO/W capacitors

Figure 1a shows typical polarization versus voltage loops measured on the W/HZO/W capacitors as a function of ac electrical cycling (the corresponding $I–V$ curves are shown in Fig. S1 of the Supplementary Information). Additional examples of $P–V$ loops upon cycling are given in the Supplementary Information. In the pristine state, a remanent polarization $P_r$ of 19.0 $\mu C/cm^2$ is observed, and the hysteresis loop is closed and saturated (whereas pinched hysteresis loops are frequent in pristine HZO capacitors with TiN electrodes). Continuing ac cycling results in a further increase of the remanent polarization, which reaches 21.5 $\mu C/cm^2$ after 2000 cycles and 22.2 $\mu C/cm^2$ after 30,000 cycles (Fig. 1b). Considering all samples/capacitors studied, $P_r$ is typically of 19–23 $\mu C/cm^2$ in the pristine state, of 22–28 $\mu C/cm^2$ after $10^4$

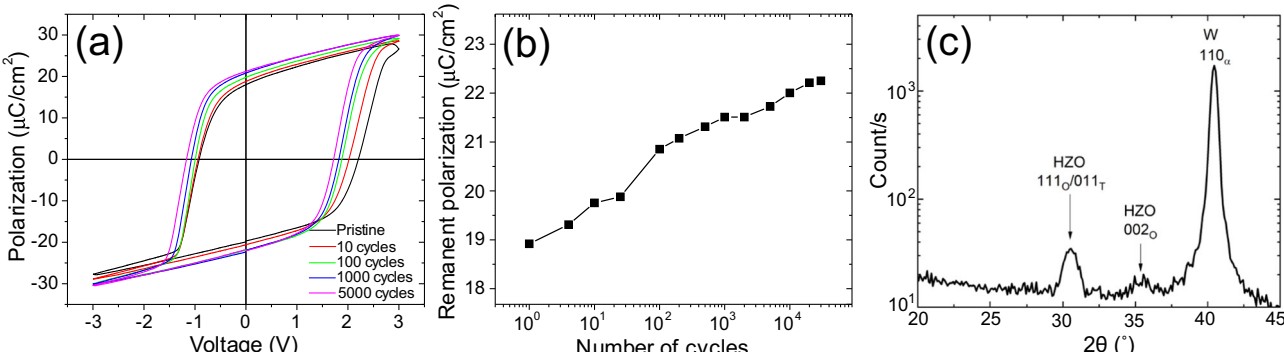

**Fig. 1 | Ferroelectric switching and X-ray diffraction data. a** Polarization hysteresis loops versus the set voltage of a W/HZO/W capacitor in the pristine state and after various numbers of ac field cycles. **b** Remanent polarization (average of $P_r^+$ and $P_r^-$) as a function of the ac cycles. **c** Grazing incidence X-ray diffraction pattern of the pristine sample (W/HZO/W with full W top sheet layer).

cycles and of $\sim 30\ \mu C/cm^2$ after $10^5$ cycles. The $P_r$ values are similar to those reported for HZO stacks with a top W electrode[45,46]. The HZO capacitors with bottom and top W electrodes are almost wake-up free, which is significantly different from other polycrystalline HfO$_2$-based ferroelectric capacitors, which require a wake-up phase of typically $\sim$ 1000–2000 field cycles to reach open saturated loops and close to full remanent polarization[11,17,34,47–49]. Wake-up-free behavior with W top and bottom electrodes has been previously reported[46]. The fact that $P_r$ in the pristine state is already quite large indicates that the pristine film contains a large portion of polar phase. From the X-ray diffraction patterns of the pristine samples, no monoclinic phase is observed (Fig. 1c). A typical peak is recorded at $2\theta \sim 30.5°$, which can be assigned to the tetragonal and/or polar o-III orthorhombic phase. Since the peak width depends on the grain size and on the microstrain in the film, it is not possible to unambiguously assign this peak to one phase (o-III) or to a mixture of t and o-III phases. We also observe the 002 peak of the o-III phase.

### Inversion of the net piezoresponse signal upon ac cycling

Figure 2 shows a series of PFM phase images recorded in the same location of a poled W/HZO/W capacitor at different stages of the ac training process. The measurement conditions are specified in the Methods section. After the application of a certain number of the field cycles, the capacitors were poled by the application of positive or negative voltage pulses with an amplitude well above the coercive voltage. The PFM images of the pristine capacitor (Fig. 2a, b) reveal uniform contrast after poling, indicating a fully switchable polarization. This result is consistent with the large remanent polarization in the pristine state observed in the macroscopic $P$–$V$ loop measurement (Fig. 1a, black curve). Analysis of the calibrated PFM phase signal[42] of the capacitor poled to the downward polarization state (appearing with dark contrast in the PFM image (Fig. 2a)) suggests that it oscillates in-phase with the ac drive signal (i.e., the capacitor expands under a positive bias). Correspondingly, the oscillation of the pristine capacitor poled upward (appearing with bright contrast in PFM (Fig. 2b)) is out-of-phase with the ac drive signal (the capacitor contracts under a positive bias). This is an indication of a positive, effective piezocoefficient $d_{33}$ in the pristine HZO. However, ac cycling leads to the appearance of regions with an inverted PFM response after poling: these regions exhibit an out-of-phase PFM signal (bright contrast) after positive pulse application and, correspondingly, in-phase signal (dark contrast) after a negative pulse (Fig. 2c,d). Although the emergence of the regions with an inverted PFM response appears as polarization switching against the applied field (the so-called "anomalous" switching), PFM phase signal analysis gives a very clear indication that these regions exhibit a negative $d_{33}$ piezocoefficient. The area occupied by the regions with negative $d_{33}$ continuously increases as a function of ac

training (Fig. 2e–j). After 5000 cycles, more than 95% of the capacitor area shows an inverted PFM phase contrast in comparison to the pristine state (Fig. 2k, l). Full $d_{33}$ inversion requires typically 1000–5000 cycles, depending on the capacitor. The net zero piezoresponse of the whole capacitor (Fig. 2m and Fig. S2 of the Supplementary Information) results from the coexistence of positive and negative $d_{33}$ regions. It is remarkable to note that the capacitor keeps a robust and fully switchable polarization (Fig. 1b and Fig. S2g) when its net piezoresponse is null.

### Inversion of the local piezoresponse signal upon ac cycling

To further explore the evolution of the piezoresponse with ac training, local PFM switching spectroscopy loops were collected at several locations. Figure 3a–f shows the evolution of the representative biasoff PFM loops with an increasing number of ac switching cycles. In the pristine state, the PFM testing reveals a conventional hysteretic loop corresponding to a positive piezoelectric coefficient (Fig. 3a). The PFM signal decreases in magnitude after 25 cycles of ac training (Fig. 3b) and after 100 cycles the PFM hysteresis loop appears as mirror-reflected indicating an inversion of the PFM signal, which is still of low magnitude (Fig. 3c). Further ac training increases the PFM signal magnitude and eventually results in a hysteretic loop corresponding to the negative piezoelectric coefficient (Fig. 3d–f). This gradual change of the PFM spectroscopy loop as a function of ac training cycles suggests that the inversion of the $d_{33}$ coefficient at the local level is not instantaneous but rather gradual: from a positive $d_{33}$ through a continuous reduction of its magnitude going through zero, followed by its sign inversion and then a gradual increase of the negative $d_{33}$ value. This is illustrated in Fig. 3g for two different locations of a capacitor measured after a very progressive ac cycling to follow closely the evolution of the electromechanical response (the piezoresponse amplitude and phases are shown in Fig. S3 of the Supplementary Information). Hence, not only can the net piezoresponse of the whole capacitor be nullified, but also the local piezoresponse. Depending on the location of the capacitor, the nullification of the electromechanical response occurs for an ac cycling of tens to thousands of cycles. The $P$–$V$ loops measured after each applied ac cycle show a non-zero polarization of the capacitor. While we cannot ascertain the existence of a switchable polarization at the local scale when the local $d_{33}$ is nullified, it is most plausible to assume that the local polarization does not suddenly go to zero while being a robust one ($>20\ \mu C/cm^2$) just before and after the nullification. Note that the tip-sample contact radius is estimated to not exceed 15 nm, which implies that a maximum of 2–3 grains are involved in each of the local switching spectroscopy measurements.

To check if a local back-switching effect (for example, due to the presence of an internal built-in field) takes place upon removal of the

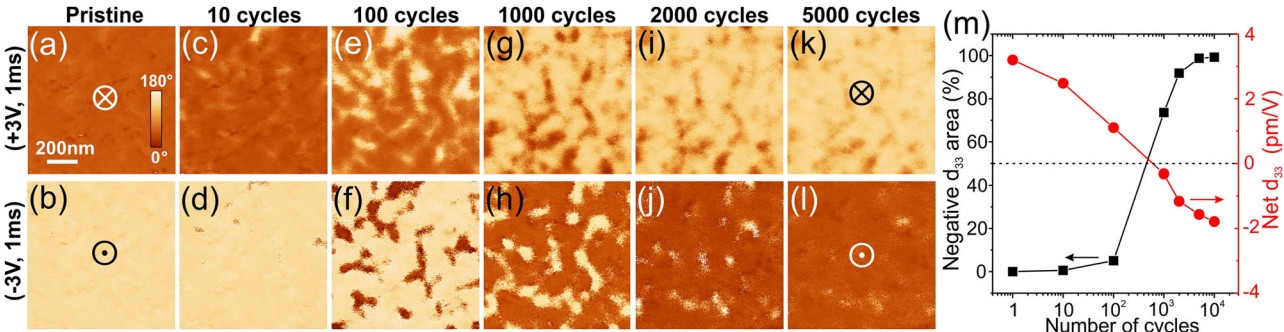

**Fig. 2 | PFM imaging of a poled W/HZO/W capacitor as a function of the ac cycling. a–l** PFM phase images representing the domain structures after application of the positive (top row) and negative (bottom row) poling pulses collected in the same location: (**a, b**) pristine state, (**c, d**) after 10 cycles, (**e, f**) 100 cycles, (**g, h**) 1000 cycles, (**l, j**) 2000 cycles, and (**k, l**) 5000 cycles. Pulse amplitude is ±3 V, and duration is 1 ms. **m** Percentage of the area with negative piezocoefficient $d_{33}$ and the average $d_{33}$ value of the entire scanned area as a function of ac cycling obtained from image analysis of data in (**a, c, e, g, l, k**).

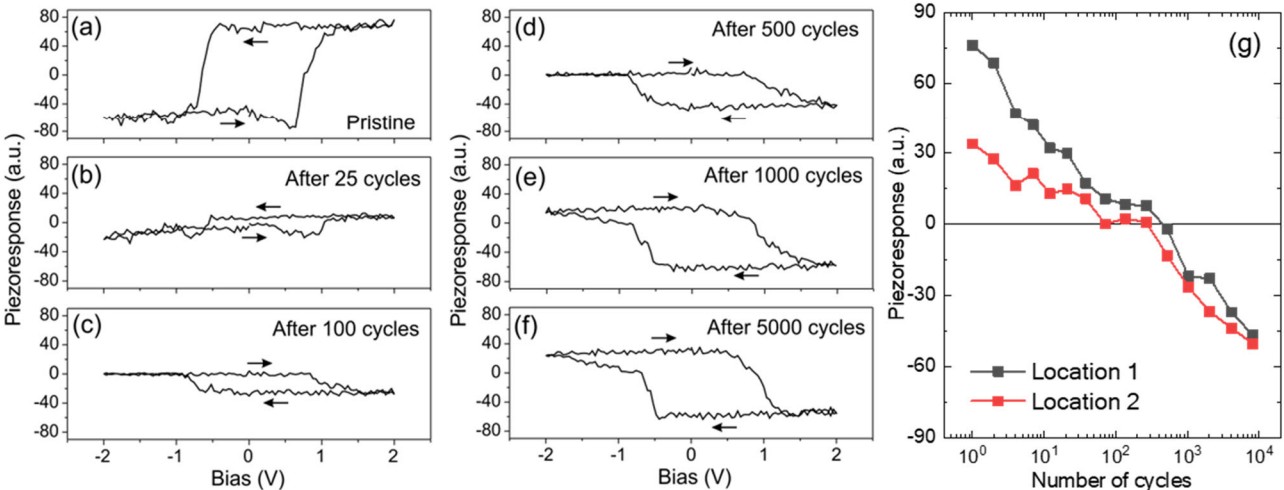

**Fig. 3 | PFM switching spectroscopy at a fixed location on a W/HZO/W capacitor as a function of the ac cycling process. a–f** Bias-off piezoresponse loops in the pristine state (**a**), after 25 cycles (**b**), 100 cycles (**c**), 500 cycles (**d**), 1000 cycles (**e**), and 5000 cycles of ac cycling (**f**), respectively. The piezoresponse loops were obtained by convoluting the PFM amplitude signal with the cosine of the PFM phase signal. **g** Local PFM amplitude evolution as a function of the number of ac cycles at two different locations of a capacitor (the bias-off PFM amplitude and phase loops are shown in Fig. S3 of the Supplementary Information).

dc poling voltages, the PFM switching spectroscopy loops were also collected simultaneously in the bias-on regime, where the piezoresponse signal was recorded in the presence of the dc bias (Fig. S4 of the Supplementary Information). The results are qualitatively similar to those shown in Fig. 3. The obtained data suggest that back-switching can be ruled out as a possible cause of the observed PFM phase inversion.

### Origin of the piezoelectric coefficient sign inversion and cancellation

Let us now discuss the possible mechanism at the origin of the inversion of the piezoelectric coefficient sign in the ferroelectric HZO upon electrical cycling. The full inversion of the $d_{33}$ sign (on a $1 \times 1\,\mu m^2$ area) is observed typically after 2000 ac cycles (we give additional examples for a different sample in Fig. S3 of the Supplementary Information). The W/HZO/W capacitors just after one ac cycle show closed saturated hysteresis loops. After 2000 cycles, the remanent polarization $P_r$ increases typically by 5–25% and keeps slowly increasing up to at least 10,000 cycles while full inversion of $d_{33}$ in the capacitor has already occurred (Fig. S3B, C). The $d_{33}$ inversion does not seem to be related to the standard wake-up mechanisms[11,49]. Extrinsic phenomena invoked[18,50] such as charge detrapping or charge redistribution, decrease of depolarization field, and domain wall depinning, cannot account for a change of the sign of $d_{33}$. Moreover, the careful calibration based on the procedure described in[42] allows us to rule out or account for the parasitic effects of any extrinsic phenomena on the measured electromechanical response. The influence of imprint on the piezoresponse could be questioned. Although in Fig. 1 we indeed observe a reduction of the coercive voltage $V_c^+$ by $\sim 0.5\,V$ after 5000 ac cycles, the vast majority of our capacitors (tested on different samples) exhibit, however, negligible change of the imprint, as we show in Fig. S5. The coercive voltage $V_c^+$ increases typically by 0.2 V over the first three to ten cycles and then remains constant (Fig. S5). The imprint window varies by less than 0.1 V upon the 10,000 cycles. Hence, we can confidently exclude that phenomena related to imprint (such as charge redistribution) account for the sign change of $d_{33}$. Electrical cycling during the wake-up phase has also been suggested to trigger a crystalline phase transition from the non-ferroelectric monoclinic or tetragonal $P4_2/nmc$ to the polar orthorhombic $Pca2_1$ phase via oxygen vacancy migration[11,51,52]. Here, we can rule out that the capacitors contain a majority of non-ferroelectric phase(s). No

monoclinic phase is observed by X-ray diffraction, and in view of the high remanent polarization $P_r$ (>20 $\mu C/cm^2$) reached in the pristine state or just after 1 cycle, we can rule out that the capacitors contain a majority of tetragonal phase. At any rate, note that a high volume fraction of the centrosymmetric ($d_{33} = 0$) tetragonal or monoclinic phases in the pristine samples would result in a weaker piezoelectric effect but cannot yield a change of sign.

We now resort to first-principles simulations to discuss whether it is possible to explain the observed sign reversal of $d_{33}$ by an *intrinsic* physical mechanism, i.e., present in a perfect HZO lattice, not related to defects or interfacial phenomena. As previously discussed in the introduction, the piezoelectric response of the usual ferroelectric phase of HfO₂ (orthorhombic, $Pca2_1$ space group) is qualitatively different from the response of the best-known piezoelectrics (i.e., perovskite oxides like PbTiO₃ or PbZr$_{1-x}$Ti$_x$O₃)[27]. In particular, the longitudinal piezoresponse is negative in HfO₂, implying that compression along the polar direction results in an enhancement of the polarization; indeed, we compute a negative $e_{33} = -144\ \mu C/cm^2$. It has been shown that this peculiar behavior stems from the chemical environment of the active oxygens that drive the occurrence of spontaneous polarization. In this work, we have run a similar DFT investigation for HZO as well as pure ZrO₂, both considered in a $Pca2_1$ configuration analogous to HfO₂'s ferroelectric state. For HZO, here we present results for the particular Hf/Zr ordering shown in Fig. S6a of the Supplementary Information (we considered alternative Hf/Zr arrangements—also shown in Fig. S6—and explicitly checked that the dependence of $e_{33}$ on the cation order is negligible (< 2%)). We obtain a theoretical spontaneous polarization of about 55 $\mu C/cm^2$ for both materials, essentially identical to the result for HfO₂. As for $e_{33}$, we obtain $-159\,\mu C/cm^2$ for HZO and $-174\,\mu C/cm^2$ for ZrO₂, confirming that the negative linear piezoresponse is a general feature of this polar state, the magnitude of the effect being weakly dependent on composition. Thus, our DFT calculations are compatible with the negative $d_{33}$ observed in our samples upon cycling.

What can then be the origin of the positive $d_{33}$ measured in the pristine samples? Our simulations suggest two possible explanations based on intrinsic mechanisms. On the one hand, we find that in these materials the sign of the longitudinal piezoresponse depends strongly on the extent to which the spontaneous polarization is developed. Our key first-principles results, summarized in Fig. 4, are obtained in the following way: we consider the paraelectric t-phase as a reference

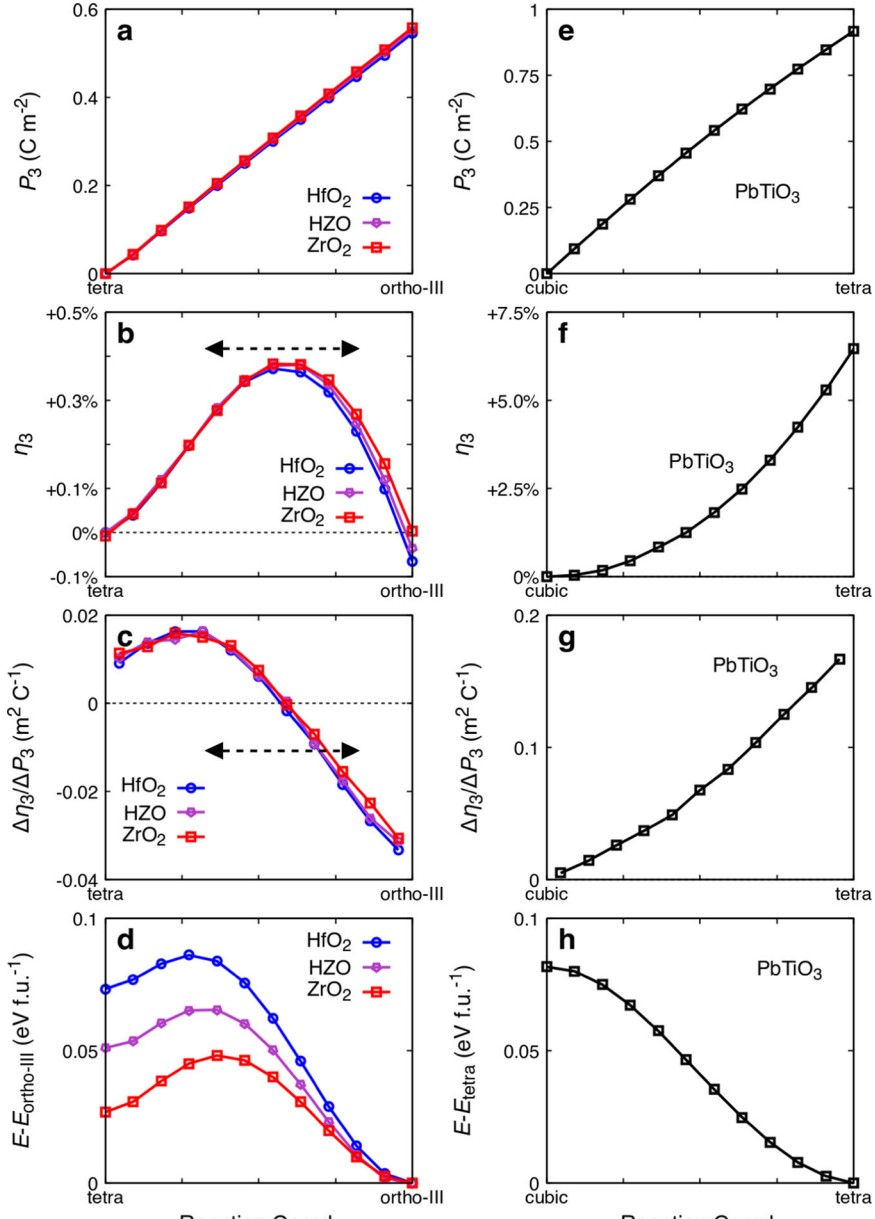

**Fig. 4 | Predicted nontrivial piezoresponse of hafnia ferroelectrics.** Connection between polarization $P_3$ and strain along the polar axis $\eta_3$, as computed for HfO$_2$, HZO, and ZrO$_2$ (**a–d**) and PbTiO$_3$ (**e–h**). The "Reaction Coordinate" in the horizontal axis refers to a linear structural interpolation between a paraelectric reference state (tetragonal for HfO$_2$, HZO, and ZrO$_2$; cubic for PbTiO$_3$) and the ferroelectric state (orthorhombic $Pca2_1$ o-III phase for HfO$_2$, HZO and ZrO$_2$; tetragonal for PbTiO$_3$). **a**, **e** show the polarization of the different materials, while **b** and **f** show the corresponding evolution of $\eta_3$. **c**, **g** show the ratio $\Delta\eta_3/\Delta P_3$ along the reaction coordinate, which roughly goes as the $d_{33}$ piezoelectric coefficient (in essence,

$d_{33} \sim \epsilon_0\chi_{33}\Delta\eta_3/\Delta P_3$, where $\epsilon_0$ is the vacuum's permittivity and $\chi_{33}$ the relevant dielectric susceptibility). **d**, **h** show the energy of the different phases as a function of the reaction coordinate, taking the polar minimum as the zero of energy. Noting the simplifications implicit in our simulations, in **b**, **c** we mark with dashed double arrows the approximate region where we expect $d_{33}$ to change sign. Interestingly, this range roughly overlaps with the one defined by the smallest ($\sim 19$–$23\ \mu C/cm^2$ in the pristine state) and largest ($30\ \mu C/cm^2$ after $10^5$ cycles) remnant polarizations experimentally measured in our samples.

structure, and simulate several configurations that can be viewed as a structural interpolation between this non-polar state and the o-III ferroelectric phase. Such intermediate configurations are constructed by displacing the atoms from the reference state as required to reach the polar configuration; the relative positions of the atoms are then fixed, and the cell strains (lattice vectors) are optimized so they can adapt to the progressive development of the orthorhombic $Pca2_1$ (polar) distortion. Thus, while the evolution of the polarization $P_3$ (Fig. 4a) is imposed by hand in these calculations, the cell strain along the polar axis, $\eta_3$ (Fig. 4b), is a result of the DFT optimization. Surprisingly, we find that $\eta_3$ does not follow the polarization monotonically; instead,

the strain grows when the polarization starts condensing, reaches a maximum (when about 60% of the full theoretical polarization has developed), and decays beyond that point. Since $d_{33}$ is essentially given by the ratio of the strain and polarization changes (Fig. 4c), our results suggest that this coefficient will reverse its sign, from positive to negative, in a continuous way as the distortion develops. Note that this is a unique feature of HfO$_2$ and related materials: our analogous results for PbTiO$_3$ (Fig. 4e–h) illustrate the behavior that is typical of ferroelectric perovskites, namely, a monotonic development of $\eta_3$ with $P_3$, yielding $d_{33} > 0$ throughout the condensation of the polar distortion. The peculiarity of HfO$_2$-related compounds is further reflected in

the complex energy landscape (Fig. 4d) connecting the paraelectric and ferroelectric ortho-III phases, which contrasts with the simplicity we find in $PbTiO_3$ (Fig. 4h).

From Fig. 4a, b, we find that the calculated maximum of $\eta_3$ for HZO corresponds to a polarization of ~33 $\mu C/cm^2$, which nearly falls within the range of polarization values measured in our capacitors in their pristine state (typically between 19 and 23 $\mu C/cm^2$), and after $10^5$ cycles (30 $\mu C/cm^2$). Note that, given the structural complexity of the HZO samples (grains, grain boundaries, phase coexistence, point and linear defects) we can hardly expect a good quantitative agreement in polarization between experiment and theory when simulating an ideal system. Hence, our simulations suggest that our pristine capacitors should present a positive $d_{33}$; further, our calculations indicate that, upon cycling, $d_{33}$ can be expected to decrease in magnitude and even become negative. Hence, notwithstanding the differences between our ideal simulated crystals and the actual polycrystalline samples, this unique predicted behavior of $HfO_2$, HZO and $ZrO_2$—i.e., the non-monotonic relationship between polarization and strain—provides a plausible intrinsic explanation to the surprising sign inversion of $d_{33}$ observed experimentally. The evolution along the calculated path under ac cycling can be attributed to the progressive development of the polar distortion of the o-III phase. The calculations remarkably account not only for the inversion of the $d_{33}$ sign but also for its behavior as observed in switching spectroscopy (Fig. 3): a continuous decrease in magnitude, going through zero, followed—after its sign inversion—by a gradual increase.

On the other hand, $HfO_2$-related compounds present many polymorphs[12], which suggests the following alternative scenario: our HZO samples might be grown in a different ferroelectric structure (with $d_{33} > 0$), then transform into the commonly assumed orthorhombic $Pca2_1$ polymorph (with $d_{33} < 0$) upon cycling. In particular, the theoretical literature on $HfO_2$ has discussed a second ferroelectric orthorhombic phase (noted "ortho-IV" in the following, with $Pmn2_1$ space group), which would be a potential candidate to appear in our as-grown films[12,53]. Both orthorhombic $HfO_2$ polar phases have been found to have similar spontaneous polarizations ($P_{Pca2_1} = 52\,\mu C/cm^2$ and $P_{Pmn2_1} = 56\,\mu C/cm^2$)[12]. We have run a standard DFT characterization of HZO in this ortho-IV structure and found that it has a spontaneous polarization of about $60\,\mu C/cm^2$ and a positive longitudinal piezoresponse $e_{33} = 100\,\mu C/cm^2$ (the atomistic mechanism for this response is described in Fig. S7 of the Supplementary Information). Hence, a transformation from ortho-IV to ortho-III upon cycling would explain the observed sign reversal in $d_{33}$. Admittedly, this interpretation is not free of problems. From a theoretical perspective, there are arguments against the possible occurrence of this phase—it has a relatively high bulk energy[12]—but also supporting it—a low surface energy[54]. Then, as far as we know, the ortho-IV phase has never been observed experimentally. Hence, we find the first proposed scenario (continuous evolution between a weakly developed and a well-developed orthorhombic polar phase) more convincing.

Additionally, let us note that it has been recently proposed[55] that the peculiar behavior of $d_{33}$ observed in $HfO_2$ (and related compounds) might be related to an intrinsic difficulty in determining the sign of the spontaneous polarization, ultimately caused by the availability of multiple ferroelectric switching paths in these materials. It is unclear whether these ideas may help explain our experiments (for example, they might imply that, upon electric cycling, the dominant switching path in our samples might be changing). Nevertheless, this reinforces the notion that exceptionality of the $HfO_2$-based ferroelectrics offers plenty of possibilities for an intrinsic explanation of our experimental results for $d_{33}$. Let us also stress that none of these scenarios apply to well-known ferroelectrics such as perovskite oxides.

## Discussion

While in the literature, some samples exhibit positive $d_{33}$ and other samples exhibit negative $d_{33}$, we experimentally showed that a complete uniform inversion of the longitudinal piezoelectric coefficient sign can be achieved in the very same $W/Hf_{0.5}Zr_{0.5}O_2/W$ ferroelectric capacitor by an electrical ac cycling process. Local PFM switching spectroscopy measurements reveal an evolution, upon ac cycling, from a positive $d_{33}$ value through a continuous reduction of its magnitude down to zero, followed by the $d_{33}$ sign inversion and a gradual increase of the negative $d_{33}$ magnitude. Concomitantly, while the polarization in the pristine state of the capacitors is already relatively large, it keeps increasing well beyond the number of cycles required to uniformly invert the sign of the piezoresponse. Our DFT calculations suggest that a progressive transformation from a weakly developed polar phase to a well-developed o-III phase provides a potential explanation for these observations. Within this model, the non-monotonic development of $\eta_3$ with $P_3$ leads to the unusual behavior of $d_{33}$, whereby the piezoresponse does not grow with $P_3$ contrary to what happens in perovskite ferroelectrics. The present results clearly resonate with previous predictions[27] that the sign of the piezoresponse of these materials can be tuned, emphasizing that electrical cycling—by controlling the development of the polar distortion—is a key knob for engineering the piezoelectric response of $Hf_{0.5}Zr_{0.5}O_2$ devices. A fine tuning of intermediate states, all the way from a positive ($\sim 3$ pm/V) to negative ($\sim -2$ pm/V) piezoresponse—including a net null one!—can be achieved with ac training while keeping a robust remanent polarization. A next challenge is to find an experimental approach (for example through annealing or mechanical strain) to reversibly change $d_{33}$ back to a positive value. For this purpose, elucidating the role of oxygen ions and oxygen vacancies movement during ac electrical cycling and understanding the interplay between ferroelectricity and ferroionicity would be key[56,57]. Further investigations are also needed to understand the specific role that the tungsten electrodes seem to play, as the full and uniform $d_{33}$ inversion in the HZO capacitors with TiN electrodes could not be realized so far (although these capacitors exhibit a similar trend, they undergo electrical breakdown before full inversion is reached). Mechanical strain imparted by electrodes (but also by processing conditions, annealing, doping etc.) is a major ingredient for the stabilization of a polar HZO phase and for determining its state evolution towards the fully developed o-III phase (for which a maximum polarization of more than $50\,\mu C/cm^2$ is predicted by DFT but not yet achieved in polycrystalline films).

With this work, we thus hope to clear the current mystery regarding the sign of piezoelectricity in the $HfO_2$-based ferroelectrics by showing that it is not an invariable parameter but a dynamic entity that can be changed in the very same material by an electric field and possibly by other external stimuli. This work also points for the first time to the possible occurrence of an intrinsic non-piezoelectric ferroelectric compound. These findings offer a fantastic playground and unprecedented perspectives to engineer the electromechanical functionality of $HfO_2$-based devices.

## Methods

### Sample and device preparation

Capacitive W/HZO/W stacks were prepared on highly doped $p^{++}$ Si substrates. First, a 30 nm-thick W layer was deposited on the silicon substrate by sputtering at room temperature. Then, a 10-nm-thick $Hf_{0.5}Zr_{0.5}O_2$ (HZO) film was grown by atomic layer deposition at 250 °C in a FlexAl OXFORD cluster tool using $Hf(NCH_3C_2H_5)_4$ (TEMAH) and $Zr(NCH_3C_2H_5)_4$ (TEMAZ) as Hf and Zr precursors respectively and $H_2O$ as an oxidizing agent. HZO was deposited by alternating in a 1:1 ratio the $HfO_2$ and $ZrO_2$ cycles except for the two first- and two last cycles, which consisted of $HfO_2$. The growth per cycle for this process was 0.15 nm/cycle. The top 25 nm-thick W

electrodes were prepared in the form of square pads of $95 \times 95\ \mu m^2$ area using a lift-off process with either standard contact photolithography or direct laser lithography and sputtering deposition at room temperature. The capacitive stacks were then annealed using a rapid thermal process at 400 °C for 120 s or 180 s under $N_2$ to crystallize the amorphous HZO films. The low-temperature annealing conditions were chosen for the compatibility of our process with back-end-of-the-line integration of the devices on CMOS chips[34]. Grazing incidence X-ray diffraction was performed on the pristine stacks with a non-patterned W layer on top (full sheet W deposition and subsequent RTA as similarly described) using Cu Kα radiation and an incident angle of 0.5 deg.

A total of five different samples with a minimum of six capacitors per sample were characterized.

## Macroscopic electrical measurements

Macroscopic polarization-voltage (*P*–*V*) loops were measured in the W/HZO/W capacitors using an arbitrary waveform generator (Keysight 33621 A) for signal generation and an oscilloscope (TDS 3014B, Tektronix) for recording the transient switching currents. A triangular waveform signal at a frequency of 10 kHz was applied to the top electrode via a micromanipulator probe in contact with the top electrode, while the bottom electrode was connected to the oscilloscope for transient current measurements. Subsequent integration of the obtained currents yielded the *P*–*V* loops. AC (sine waveform) cycling was performed with an ac voltage of 3 V at 2 kHz.

Further macroscopic *P*–*V* loop measurements were carried out using a commercial ferroelectric test system (Radiant Precision Multiferroic II). The field cycling was performed using a sine 1 kHz, ±2.8 V waveform signal, and the *P*–*V* loop measurements were performed using a triangular 1 kHz, ±3 V. For the measurements of the pristine state, we applied first a preset triangular signal.

## PFM imaging and spectroscopy

PFM measurements were performed on a commercial AFM system (MFP3D, Asylum Research) using AFM tips with single crystal diamond probes (D80, K-Tek). Dual AC resonance tracking (DART) PFM mode was used for signal enhancement, where ac drive frequencies near the AFM cantilever contact resonance (~360 kHz) and 0.5 V amplitude were applied to the top electrode via an external micromanipulator probe in contact with the electrode, and the AFM tip was used for PFM signal detection. An ac voltage with a 3 V amplitude at 2 kHz was used for ac cycling. All voltages were applied through the external tip, and the PFM tip was used only for the piezoresponse signal detection.

Additional PFM measurements were performed on the capacitors using a Park Systems NX10 microscope. A nonconducting silicon AFM tip (FMR, NanoWorld) was positioned on the top electrode and only detected the electromechanical response of the capacitor. The ac modulation for the PFM measurements was applied to the top electrode through a tungsten needle in contact with it, and other voltage applications, such as ac training cycles and square pulses, were done to the bottom electrode through the highly doped Si substrate. The ac modulation frequency was regulated at ~380 kHz by the dual frequency resonance tracking with the lock-in amplifier (UHFLI, Zurich Instruments).

With both scanning probe microscopes, the PFM measurements were performed using top metal electrodes (25 nm thick), which allows us to rule out sample/tip interaction artifacts such as electrostatic charging at the sample surface as a major contribution.

On both set-ups, we used a careful procedure described in ref. 42 to calibrate the PFM phase and to rule out or account for the parasitic effects of any extrinsic phenomena on the measured electromechanical response.

## First-principles calculations

We perform DFT calculations using the Viena Ab initio Simulation Package (VASP)[58,59]. We employ the Perdew-Burke-Ernzerhof formulation for solids (PBEsol)[60] of the generalized gradient approximation for the exchange-correlation functional. We treat the atomic cores within the projector-augmented wave approach[61], considering the following states explicitly: 5s, 5p, 6s, and 5d for Hf; 4s 4p, 5 s, and 4d for Zr; and 2s and 2p for O. We use a plane-wave energy cut-off of 600 eV for the basis set, which we found to be enough to obtain well-converged results. The Brillouin zone integrals were computed using a $6 \times 6 \times 6$ ($8 \times 6 \times 8$) Monkhorst-Pack[62] k-point grid for the primitive cell of the o-III and tetragonal (o-IV) phases, corresponding to a simulation cell of 12 (6) atoms. The stable phases were fully relaxed until residual forces fell below 0.001 eV/Å and residual stresses below 0.01 GPa. The structures along the interpolated path were allowed to relax their lattice vectors until stresses became smaller than 0.01 GPa, keeping their relative atomic coordinates fixed. We obtain the piezoelectric tensor by means of the density functional perturbation theory implementation readily available in VASP[63].

## Data availability

The data that support the images and plots within this paper and other findings of this study are available from the corresponding authors upon reasonable request.

## Code availability

The first-principles simulations were done with VASP, which is proprietary software for which the LIST group owns a license and which can be acquired from the developers (https://vasp.at/).

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

## Acknowledgements

C.D., D.-J.K., V.D., and M.H. acknowledge funding by Horizon 2020 EU project BeFerroSynaptic (No. 871737). C.D. and D.-J.K. acknowledge the funding and cooperation from Park Systems. Dr. H. Raza from HZB is acknowledged for growing additional samples. K. S. Nair and T.L. Phan, PhD, are acknowledged for performing lithography. Work at LIST (H.A, J.Í.) was funded by the Luxembourg National Research Fund through grant INTER/NWO/20/15079143/TRICOLOR. A.G. and C.D. acknowledge the support of the Alexander von Humboldt Foundation.

## Author contributions

C.D. initiated and coordinated the study. C.D. and A.G. supervised the experimental work. M.H. grew the samples and performed electrical measurements on the capacitors. H.L., with the assistance of P.B. and D.J.K., performed the PFM and macroscopic electrical measurements of the capacitors. H.A. performed the first-principles study, assisted by S.D. and supervised by J.Í. C.D., A.G., J.Í, H.L., D.J.K., H.A., M.H., P.B., S.D., U.S., and V. D. discussed the results. The manuscript was written by C.D., A.G., J.Í, D.J.K., and H.L. with contributions from H.A. All co-authors edited the manuscript.

## Funding

## Competing interests

The authors declare no competing interests.
