## [Peer Review File · Nature Communications]

Electrically-induced cancellation and inversion of piezoelectricity in ferroelectric Hf_{0.5}Zr_{0.5}O₂ capacitorsREVIEWER COMMENTS

Reviewer #1 (Remarks to the Author):

This work demonstrates that Hf_{0.5}Zr_{0.5}O₂ films subjected to ac cycling undergo a continuous transition from a positive effective piezoelectric coefficient d₃₃ to an inverted negative d₃₃ state without changing polarization switching properties based on PFM characterization. While some of the presented results (inversion of sign of piezoelectric coefficient) might be of interest to the readers of Nature Communications, overall I do not feel that the results are with rigorous investigation and are novel and significant enough to warrant publication in such a high impact journal (see details below).

1. This paper only uses PFM characterization on only one sample to get the conclusion. But it is well known that the PFM results could be interfered by many extrinsic factors and very rich experience is needed to get reliable results. Without other methods and other samples to double confirm the results, we are not sure how believable the conclusion is. For example, how about the results with thicker HZO film?

2. Particularly, the author stated that “extrinsic phenomena invoked in ac cycling such as charge de-trapping or charge redistribution, decrease of depolarization field, and domain wall depinning cannot account for a change of the sign of d₃₃.” Do these effects affect PFM characterization?

3. In the section to of “origin of the piezoelectric coefficient sign inversion and cancellation”, the author stated that “No monoclinic phase is observed by X-ray diffraction, and in view of the high remanent polarization Pr (> 20 μC/cm²) reached in the pristine state or just after 1 cycle, we can rule out that the capacitors contain a majority of tetragonal phase.” It is understandable that M-phase can be rule out, but how the T-phase is excluded simply as T-phase coexists with O-phase generally even for a high Pr film.

4. In the conclusion, the paper stated that “the current controversy regarding the sign of piezoelectricity in the HfO₂-based ferroelectrics by showing that it is not an invariable parameter ...” But in the introduction, it has been also said “it has been observed that the sign of the d₃₃ could vary depending on the film thickness, electrode materials and deposition method used [38-42]. In addition, coexistence of regions with positive and negative d₃₃ piezocoefficients within the same device has been reported 78 [42].” It seems the previous work has demonstrated the results that the piezoelectricity is not an invariable parameter. So, we are not sure how significant the conclusion is even if it is confirmable.

5. There are some writing issues. For example, the forms of unit are not unified (μC/cm² and μC cm⁻²) and the unit of d₃₃ in line 311 is wrong.

Reviewer #2 (Remarks to the Author):

I have carefully read the manuscript entitled “Electrically-induced cancellation and inversion of piezoelectricity in ferroelectric Hf_{0.5}Zr_{0.5}O₂ capacitors”. The authors show that piezoelectric coefficient in hafnia-based capacitors change from positive to negative while cycling. The polarization, in contrast suffers from a small change. The change of sign of piezoelectric response while cycling is of interest from fundamental point of view and for applications.

The authors show that the polarization evolution while cycling is completely disconnected from the piezoelectric response evolution. It would help reader understanding add clear statements regarding this surprising fact in abstract/introduction/conclusions.

Figure 1a reveals that there is a reduction of imprint electric field while cycling. This fact results in a reduction of V_{c+}. This latter fact might result in a “better” saturation at the used

maximum applied electric field. Thus, imprint reduces while cycling and as a result polarization increases. One can argue that this experimental fact can have an impact on the measured piezoresponse. The presence of dead layer with charged vacancies and positive piezoelectric response that reduces while cycling because of defects redistribution is probably accounting for both effects. When dead-layer reduces upon cycling intrinsic negative bulk intrinsic piezoelectric response emerges. Comment about the imprint evolution with cycling and its possible connection with the piezoelectric response sign reversal seems necessary.

Authors mention “After the application of a certain number of the field cycles, the capacitors were poled by application of positive or negative voltage pulses with an amplitude well above the coercive voltage.”. The final voltage pulse is in-situ applied with AFM tip or ex-situ? Are top and bottom images of Figure 2 are collected in the same region. Clarify in the text.

Figure 2 shows that overall change is gradual. Figure 3 and discussion show a local gradual change. Figure 2 shows that for each pixel of the image the change is not gradual, i.e. all the images show similar two colors. Therefore, at local scale there is an apparent discrepancy. From result show in Figure 3, one expects that the intensity of the images of Figure 2 should be different after different number of cycles. This might be related to the fact that panels in Figure 2 are not all at the same scale. In any case, this apparent discrepancy must be clarified.

The authors should detail more the following argument: pg 9. L241 “we checked that this choice does not have a significant influence on our properties of interest”

It seems necessary to cite the following works to put in better context the results:

- Lomenzo, P.D., Collins, L., Ganser, R., Xu, B., Guido, R., Gruverman, A., Kersch, A., Mikolajick, T. and Schroeder, U. (2023), Discovery of Nanoscale Electric Field-Induced Phase Transitions in ZrO₂. *Adv. Funct. Mater.* 2303636.

<https://doi.org/10.1002/adfm.202303636>

- Kelley, K.P., Morozovska, A.N., Eliseev, E.A. et al. Ferroelectricity in hafnia controlled via surface electrochemical state. *Nat. Mater.* (2023). <https://doi.org/10.1038/s41563-023-01619-9>

Reviewer #3 (Remarks to the Author):

Ferroelectricity of HfO₂-based thin films is of great technical importance, which has not been well understood yet. And the piezoelectricity of these films has also attracted great interest. Authors for the first time find the continuous transition of effective piezoelectric coefficient d_{33} of Hf_{0.5}Zr_{0.5}O₂ films from positive to negative. Due to the authors have many experiences in this field and the different signs of d_{33} for HfO₂-based thin films have been reported already, the present measured results are sound to me. Overall, I think this paper can be publishable on Nature Communication. However, I still have one concern about the understanding of the transition of piezoelectric coefficient before the paper can be publishable.

Based on the first-principles simulations, authors raise a positive d_{33} in the weakly-developed polar orthorhombic phase (near the tetragonal paraelectric state) and a negative d_{33} in the well-developed orthorhombic polar phase (near the orthorhombic ferroelectric state). However, in the experiments, authors show that the pristine polarization is already 26 $\mu\text{C}/\text{cm}^2$ and the polarization after training process is 31 $\mu\text{C}/\text{cm}^2$. The polarization change is not so obvious as that found in the literatures. This indicates that the difference between the phase compositions before and after wake-up process may be not so significant. And the

measured polarizations are all less than the $33 \mu\text{C}/\text{cm}^2$ for the maximum strain along the polar axis. Can the authors have some more discussion about this?

Response to the Reviewers Comments

We thank the reviewers for the comments and questions.

Reviewer #1 (Remarks to the Author):

General Comment. This work demonstrates that $\text{Hf}_{0.5}\text{Zr}_{0.5}\text{O}_2$ films subjected to ac cycling undergo a continuous transition from a positive effective piezoelectric coefficient d_{33} to an inverted negative d_{33} state without changing polarization switching properties based on PFM characterization. While some of the presented results (inversion of sign of piezoelectric coefficient) might be of interest to the readers of Nature Communications, overall I do not feel that the results are with rigorous investigation and are novel and significant enough to warrant publication in such a high impact journal (see details below).

Response. We respectfully disagree with this reviewer's assessment regarding novelty, rigor and significance of our manuscript. The central point of the manuscript is the experimental observation of a unique phenomenon - an electrically induced change of the sign piezoelectric coefficient d_{33} in the $\text{Hf}_{0.5}\text{Zr}_{0.5}\text{O}_2$ capacitors. Such an effect has never been reported before for any piezoelectrically active material – what could be more novel than that? This experimental result is supported by the DFT calculations, which not only explain the mechanism of the d_{33} sign inversion but also predict a groundbreaking result: a possible occurrence of an intrinsic non-piezoelectric ferroelectric compound, which we observe experimentally. As for the significance of the reported finding, tunable piezoelectricity in conjunction with the robust switchable polarization create an unprecedented potential for achieving versatile electromechanical functionality of the ferroelectric HfO_2 -based devices.

Comment 1. This paper only uses PFM characterization on only one sample to get the conclusion. But it is well known that the PFM results could be interfered by many extrinsic factors and very rich experience is needed to get reliable results. Without other methods and other samples to double confirm the results, we are not sure how believable the conclusion is. For example, how about the results with thicker HZO film?

Response. We are certainly aware about potential pitfalls of PFM characterization and took special measures to eliminate any possible artifacts. The methodology for careful calibration of the PFM signal used in this study has been developed and tested by some of the co-authors of this manuscript and reported elsewhere (Buragohain *et al.*, Adv. Mat. 2206237 (2022) – ref 42 of the manuscript). This ensures that our experimental method detects a genuine artifact-free piezoelectric response of the HZO capacitors.

The obtained results are highly reproducible and contrary to this reviewer's assumption, have been observed in several samples of the same thickness and composition. They have been obtained and reproduced on two different PFM set-ups in Prof. Gruverman's group and in Prof.

Dubourdieu's group. A total of 5 samples have been investigated with a minimum of 6 devices tested on each sample (we had given in the Supplementary Information section (Fig. S2), on purpose, additional PFM images of the evolution of the phase during electrical cycling, which were measured on two devices of a different sample than the one of Fig. 2).

The reviewer's question regarding the thicker HZO films is not relevant here. We observed a new, never-before reported effect of electrically tunable piezoelectricity – this is the main point of the manuscript. It is a matter of separate studies to investigate the effect of thickness, composition, processing methods and other factors on manifestation of this phenomenon.

Manuscript modifications:

Main text, page 8:

The full inversion of the d_{33} sign (on a $1 \times 1 \mu\text{m}^2$ area) is observed typically after 2000 ac cycles (we give additional examples for a different sample in Fig. S2 of the Supplementary Information)”

Methods section, page 14:

“A total of five different samples with a minimum of six capacitors per sample were characterized.”

Comment 2. Particularly, the author stated that “extrinsic phenomena invoked in ac cycling such as charge de-trapping or charge redistribution, decrease of depolarization field, and domain wall depinning cannot account for a change of the sign of d_{33} .” Do these effects affect PFM characterization?

Response. As we mentioned in our response to Comment 1 of this reviewer, we used a carefully calibrated PFM testing procedure (described in Buragohain *et al.*, Adv. Mat. 2206237 (2022)) to rule out or account for the parasitic effect of any extrinsic phenomena on the measured electromechanical response. It is the use of this approach that ensures that we detect a genuine piezoelectric signal. We have clarified this again in the manuscript.

Manuscript modifications:

Main text, page 8:

“Moreover, the careful calibration based on the procedure described in [42] allows us to rule out or account for the parasitic effects of any extrinsic phenomena on the measured electromechanical response.”

Method section page 15:

“On both set-ups, we used a careful procedure described in [42] to calibrate the PFM phase and to rule out or account for the parasitic effects of any extrinsic phenomena on the measured electromechanical response.”

Comment 3. In the section to of “origin of the piezoelectric coefficient sign inversion and cancellation”, the author stated that “No monoclinic phase is observed by X-ray diffraction, and in view of the high remanent polarization P_r ($> 20 \mu\text{C}/\text{cm}^2$) reached in the pristine state or just after 1 cycle, we can rule out that the capacitors contain a majority of tetragonal phase.” It is understandable that M-phase can be rule out, but how the T-phase is excluded simply as T-phase coexists with O-phase generally even for a high P_r film.

Response: There seems to be a misunderstanding. We do not say that there is no T phase in the films sandwiched in-between W electrodes. We can certainly not rule out its presence and X-ray diffraction does not allow to discriminate o- and t-phases from the peak at $2\theta \sim 30.5^\circ$. What we do say is that we rule out that there is a majority of T phase. In our capacitors (with W bottom and top electrodes), the polarization after just one cycle is large, of 21-26 $\mu\text{C}/\text{cm}^2$, and the hysteresis loops are already fully sharp and closed.

Comment 4. In the conclusion, the paper stated that” the current controversy regarding the sign of piezoelectricity in the HfO_2 -based ferroelectrics by showing that it is not an invariable parameter” But in the introduction, it has been also said “it has been observed that the sign of the d_{33} could vary depending on the film thickness, electrode materials and deposition method used [38-42]. In addition, coexistence of regions with positive and negative d_{33} piezocoefficients within the same device has been reported 78 [42].” It seems the previous work has demonstrated the results that the piezoelectricity is not an invariable parameter. So, we are not sure how significant the conclusion is even if it is confirmable.

Response. We strongly disagree. In fact, it seems to us that the Reviewer may be missing the point. We are fully aware that current literature has indeed many experimental reports on different signs of piezoelectricity in different hafnia-based materials, i.e. with different stoichiometry, thickness, electrode materials, prepared by different deposition methods. Some samples exhibit positive d_{33} and other samples exhibit negative d_{33} . It has, however, never been reported that the sign of piezoelectricity can be inversed in the very same sample by an external stimulus, i.e. by electrical cycling in our case. Therefore, the reviewer’s statement that “the previous work has demonstrated the results that the piezoelectricity is not an invariable parameter” is factually incorrect.

We have revised the manuscript in an effort to explain this point in an even clearer way, so there is no room for misunderstanding.

Modification of the manuscript to emphasize this point:

Discussion, page 13:

“While in the literature some samples exhibit positive d_{33} and other samples exhibit negative d_{33} , we experimentally showed that a complete uniform inversion of the longitudinal piezoelectric coefficient sign can be achieved in the very same $\text{W}/\text{Hf}_{0.5}\text{Zr}_{0.5}\text{O}_2/\text{W}$ ferroelectric capacitor by an electrical ac cycling process.”

Since the word “controversy” in our conclusion (“the current *controversy* regarding the sign of piezoelectricity in the HfO₂-based ferroelectrics”) might be misleading, we have changed it and added one more time that the dynamic change of d_{33} is realized in the same material.

Modification of the manuscript:

End of Discussion, page 14:

“With this work, we thus hope to clear the current **mystery** regarding the sign of piezoelectricity in the HfO₂-based ferroelectrics by showing that it is not an invariable parameter but a dynamic entity that can be changed **in the very same material** by an electric field and possibly by other external stimuli.”

Comment 5. There are some writing issues. For example, the forms of unit are not unified ($\mu\text{C}/\text{cm}^2$ and $\mu\text{C cm}^{-2}$) and the unit of d_{33} in line 311 is wrong.

Response: We have unified the unit and replaced all $\mu\text{C.cm}^{-2}$ by $\mu\text{C}/\text{cm}^2$. As for the value of d_{33} in line 311: this is a result for e_{33} , not d_{33} . The typo has been corrected.

Reviewer #2 (Remarks to the Author):

General Comment. I have carefully read the manuscript entitled “Electrically-induced cancellation and inversion of piezoelectricity in ferroelectric Hf_{0.5}Zr_{0.5}O₂ capacitors”. The authors show that piezoelectric coefficient in hafnia-based capacitors change from positive to negative while cycling. The polarization, in contrast suffers from a small change. The change of sign of piezoelectric response while cycling is of interest from fundamental point of view and for applications.

The authors show that the polarization evolution while cycling is completely disconnected from the piezoelectric response evolution. It would help reader understanding add clear statements regarding this surprising fact in abstract/introduction/conclusions.

Response Indeed, this point should be more clearly emphasized. We have added sentences (in the abstract, introduction, main text and discussion) to point out indeed to the disconnected, smooth increase of the polarization while d_{33} changes sign upon ac cycling.

Modifications of the manuscript:

Abstract: line 8:

“while the polarization evolution appears completely disconnected from the d_{33} evolution.”

Introduction (at the end), page 4:

“They account for the surprising experimental observation that the polarization evolution upon ac cycling is completely decoupled from the piezoelectric response evolution.”

Main text (section “Inversion of the net piezoresponse signal upon ac cycling”), page 6:

“It is remarkable to note that the capacitor keeps a robust and fully switchable polarization (Fig. 1b and Fig. S2g) when its net piezoresponse is null indicating that the polarization evolution is decoupled from the piezoresponse evolution.”

Discussion, page 13:

“Within this model, the polarization evolution appears indeed disconnected from the piezoelectric response evolution contrarily to what happens in perovskite ferroelectrics.”

Comment 1. Figure 1a reveals that there is a reduction of imprint electric field while cycling. This fact results in a reduction of V_c^+ . This latter fact might result in a “better” saturation at the used maximum applied electric field. Thus, imprint reduces while cycling and as a result polarization increases. One can argue that this experimental fact can have an impact on the measured piezoresponse. The presence of dead layer with charged vacancies and positive piezoelectric response that reduces while cycling because of defects redistribution is probably accounting for both effects. When dead-layer reduces upon cycling intrinsic negative bulk intrinsic piezoelectric response emerges. Comment about the imprint evolution with cycling and its possible connection with the piezoelectric response sign reversal seems necessary.

Response. The presence of a dead layer with charged vacancies (and positive piezoelectric response) that reduces while cycling because of defects redistribution is usually mentioned as a reason for the wakeup phenomenon. In our capacitors, there is no wake up *per se* but a continuous increase of the polarization which keeps increasing well beyond the number of cycles needed to reverse the sign of d_{33} (as mentioned in the Discussion).

Moreover, the vast majority of our capacitors (tested on different sample batches) exhibit a negligible change of the imprint. The coercive voltage V_c^+ increases while V_c^- decreases upon the three first cycles and then remain overall constant. We have added Fig. S4 to illustrate this point (the values are those from the P-V loops shown in Fig. S2). Hence, we can confidently exclude that phenomena related to imprint account for the sign change of d_{33} .

Modifications of the manuscript:

Main text, page 8/9:

“The influence of imprint on the piezoresponse could be questioned. Although in Fig. 1 we indeed observe a reduction of the coercive voltage V_c^+ by ~ 0.5 V after 5000 ac cycles, the vast majority of our capacitors (tested on different samples) exhibit, however, negligible change of the imprint, as we show in Fig. S4. The coercive voltage V_c^+ increases typically by 0.2 V over the first three to ten cycles and then remain constant (Fig. S4). The imprint window varies by less than 0.1 V upon the 10000 cycles. Hence, we can confidently exclude that phenomena related to imprint (such as charge redistribution) account for the sign change of d_{33} .”

Supporting Information: We have added Figure S4 (showing the evolution of the coercive voltages V_c^+ , V_c^- and of the imprint). Subsequent supplementary figures have been renumbered.

Fig. S4: Imprint on two different W/HZO/W capacitors as a function of the number of ac electrical cycles. (a) Coercive voltages V_c^+ and V_c^- extracted from the P-V loops shown in (c) of Fig. S2 for the two capacitors named L and R and imprint (calculated as $\frac{(V_c^+ + V_c^-)}{2}$), as a function of ac cycles. The imprint changes by less than 0.1 V over 10 000 cycles.

Comment 2. Authors mention “After the application of a certain number of the field cycles, the capacitors were poled by application of positive or negative voltage pulses with an amplitude well above the coercive voltage.”. The final voltage pulse is in-situ applied with AFM tip or ex-situ? Are top and bottom images of Figure 2 are collected in the same region. Clarify in the text.

Response. The capacitors were poled by application of a single poling pulse after a certain number of high-amplitude ac cycles. All voltages were applied to the top electrode via an external probe and the PFM tip was used for PFM signal detection only. Poling pulse application was followed by PFM imaging carried out in a conventional way, i.e. with only low-amplitude ac imaging bias applied. Images in Figure 2 were acquired in the same location. We have added clarification into the text.

Modifications of the manuscript:

Main text (section “Inversion of the net piezoresponse signal upon ac cycling”), page 5:

“Figure 2 shows a series of PFM phase images recorded in the same location of a poled W/HZO/W capacitor at different stages of the ac training process.”

Caption of Figure 2:

“PFM imaging of the poled W/HZO/W capacitor as a function of the ac cycling. (a-l) PFM phase images representing the domain structures after application of the positive (top row) and negative (bottom row) poling pulses collected in the same location: ...”

Method section, page 15:

“An ac voltage with a 3 V amplitude at 2 kHz was used for ac cycling. All voltages were applied through the external tip and the PFM tip was used only for the piezoresponse signal detection.”

Comment 3. Figure 2 shows that overall change is gradual. Figure 3 and discussion show a local gradual change. Figure 2 shows that for each pixel of the image the change is not gradual, i.e. all

the images show similar two colors. Therefore, at local scale there is an apparent discrepancy. From result show in Figure 3, one expects that the intensity of the images of Figure 2 should be different after different number of cycles. This might be related to the fact that panels in Figure 2 are not all at the same scale. In any case, this apparent discrepancy must be clarified.

Response. We appreciate this comment. The reason that each individual pixel in the Figure 2 images shows only two colors, is that those images represent a PFM *phase* signal, which is a binary function of the polarization state. A gradual change in piezoelectricity is reflected in the PFM amplitude signal, which is incorporated into the piezoresponse loops in Figure 3 and causes evolution of their vertical span. In the Supplementary section, we have added the PFM amplitude images corresponding to the phase images in Figure S2. Therein, it can be seen that the PFM amplitude undergoes a gradual transition from a strong signal (bright contrast) to a weak signal (dark contrast) and to the strong signal again.

Modifications of the Supplementary Section:

Figure S2: addition of the PFM amplitude images corresponding to the phase images in Fig. S2.

Comment 4. The authors should detail more the following argument: pg 9. L241 “we checked that this choice does not have a significant influence on our properties of interest”

Response. We computed the piezoelectric response of the $Pca2_1$ phase of $Hf_{0.5}Zr_{0.5}O_2$ for several arrangements of the Hf and Zr cations. More precisely, we considered the three inequivalent Hf/Zr orders shown in Fig. S5 and obtained the following values for e_{33} : $-159 \mu C/cm^2$, $-156 \mu C/cm^2$ and $-157 \mu C/cm^2$. We thus concluded that the particular Hf/Zr ordering does not have a significant impact in the piezoresponse and did not pursue this point further. In the manuscript we report the results for a Hf/Zr arrangement shown in Fig. S5 a. We have revised the text to briefly make this point.

Modifications of the manuscript:

Main text (section “Origin of the piezoelectric coefficient sign inversion and cancellation “), p. 9 “For HZO, here we present results for the particular Hf/Zr ordering shown in Fig. S5a of the Supplementary Information (we considered alternative Hf/Zr arrangements – also shown in Fig. S5 – and explicitly checked that the dependence of e_{33} on the cation order is negligible (<2%))”.

Comment 5. It seems necessary to cite the following works to put in better context the results:

- Lomenzo, P.D., Collins, L., Ganser, R., Xu, B., Guido, R., Gruverman, A., Kersch, A., Mikolajick, T. and Schroeder, U. (2023), Discovery of Nanoscale Electric Field-Induced Phase Transitions in ZrO_2 . Adv. Funct. Mater. 2303636. <https://doi.org/10.1002/adfm.202303636>
- Kelley, K.P., Morozovska, A.N., Eliseev, E.A. et al. Ferroelectricity in hafnia controlled via surface electrochemical state. Nat. Mater. (2023). <https://doi.org/10.1038/s41563-023-01619-9>

Response. We appreciate this comment. Indeed, our proposed mechanism behind the observed inversion of piezoelectricity and the recently reported field-induced phase transitions in ZrO₂ films (Lomenzo *et al.*) are likely to be related by the key role of ionic movement. Further studies are necessary to clarify the role of ionic migration recently observed in the environment-controlled PFM studies of hafnia (Kelley *et al.*). We have added these references and added a corresponding remark in the Discussion.

Modification of the manuscript:

Discussion, page 13:

“For this purpose, elucidating the role of oxygen ions and oxygen vacancies movement during electrical cycling and understanding the interplay between ferroelectricity and ferroionicity are key [56, 57].”

Addition of references 55 and 56. The subsequent references have been renumbered.

Reviewer #3 (Remarks to the Author):

Ferroelectricity of HfO₂-based thin films is of great technical importance, which has not been well understood yet. And the piezoelectricity of these films has also attracted great interest. Authors for the first time find the continuous transition of effective piezoelectric coefficient d_{33} of Hf_{0.5}Zr_{0.5}O₂ films from positive to negative. Due to the authors have many experiences in this field and the different signs of d_{33} for HfO₂-based thin films have been reported already, the present measured results are sound to me. Overall, I think this paper can be publishable on Nature Communication. However, I still have one concern about the understanding of the transition of piezoelectric coefficient before the paper can be publishable.

Based on the first-principles simulations, authors raise a positive d_{33} in the weakly-developed polar orthorhombic phase (near the tetragonal paraelectric state) and a negative d_{33} in the well-developed orthorhombic polar phase (near the orthorhombic ferroelectric state). However, in the experiments, authors show that the pristine polarization is already 26 $\mu\text{C}/\text{cm}^2$ and the polarization after training process is 31 $\mu\text{C}/\text{cm}^2$. The polarization change is not so obvious as that found in the literatures. This indicates that the difference between the phase compositions before and after wake-up process may be not so significant. And the measured polarizations are all less than the 33 $\mu\text{C}/\text{cm}^2$ for the maximum strain along the polar axis. Can the authors have some more discussion about this?

Response: We agree with the Reviewer that, in these samples, the polarization change induced by the electrical cycling is relatively small (i.e., the samples are “awake” from the start), which seems to suggest that this polarization variation is unlikely to explain the change of sign in d_{33} . Note that other authors have reported such “absence” of wake-up in W/HZO/W capacitors as we

mention in the text, page 4 (“The P_r values are similar to those reported for HZO stacks with a top W electrode [45, 46]. The HZO capacitors with bottom and top W electrodes are almost wake-up free, which is significantly different from other polycrystalline HfO_2 -based ferroelectric capacitors, which require a wake-up phase of typically $\sim 1000 - 2000$ field cycles to reach open saturated loops and close to full remanent polarization [11, 17, 34, 47-49]. Wake-up free behavior with W top and bottom electrodes has been previously reported [46].”).

However, the first-principles calculations (Fig. 4 of the manuscript) clearly indicate that d_{33} will change sign for polarization values around $30 \mu\text{C}/\text{cm}^2$. Naturally, given the structural complexity of our actual samples – and the simplicity of our simulated HZO – we cannot expect a precise correspondence between this theoretical estimate of $30 \mu\text{C}/\text{cm}^2$ and the actual conditions at which d_{33} will change sign in the experiments. Yet, a transition from positive to negative d_{33} is clearly expected (this already makes HZO unique!), and it is expected for polarization values that are not far from those measured in our samples. Moreover, the change in d_{33} sign is somewhat “disconnected” from the polarization evolution which keeps increasing. Hence, it is plausible that our experimental observations reflect this predicted behavior.

We should note here that the differences between simulated HfO_2 (or HZO) and the actual samples are well-known. For example, and most relevant to this discussion, the theoretical polarization of these materials – as predicted from accurate calculations based on Density Functional Theory – is about $55 \mu\text{C}/\text{cm}^2$. By contrast, the (largest) remanent polarization values reported in the experimental literature for polycrystalline films are significantly smaller ($P_r < 40 \mu\text{C}/\text{cm}^2$). We had mentioned this fact in the Discussion, page 14 with “towards the fully developed o-III phase (for which a maximum polarization of more than $50 \mu\text{C}/\text{cm}^2$ is predicted by DFT but not yet achieved in polycrystalline films)”. Many factors may contribute to this discrepancy, stemming from the structural complexity of these compounds – in the actual samples we have grains, grain boundaries, possibly phase coexistence, point and linear defects, etc. For this reason, when it comes to quantitative comparisons between theory and experiment for HfO_2 -related ferroelectrics, we think we should moderate our expectations and, rather, focus on qualitative effects and trends. This is the viewpoint we have adopted in this article. To make this point, we have added a sentence in the manuscript.

Modifications of the manuscript:

Main text (section “Origin of the piezoelectric coefficient sign inversion and cancellation”), p. 10
“Note that, given the structural complexity of the HZO samples (grains, grain boundaries, phase coexistence, point and linear defects) we can hardly expect a good quantitative agreement in polarization between experiment and theory when simulating an ideal system.”

REVIEWER COMMENTS

Reviewer #1 (Remarks to the Author):

The authors have addressed some of the issues pointed out in last time of review. However, some issues concerning the content and response are still unclear as below.

1. In Fig. 1(a) the authors give the P-V hysteresis loops of the W/HZO/W capacitor in the pristine state and after various numbers of ac field cycles. This is a very important information for the authors' discussion in response letter. However, the authors still need to provide some additional information to make things clearer:

i) there are some abnormal zig-zag curves in Fig. 1(a). The authors should give the corresponding I-V and V-t curves to help understanding the cause of abnormal jitter.

ii) It can be easily found that the sample has around 0.5 V in-built electric field which makes the coercive field is no longer symmetric about 0. This also makes the forward switching, especially on the first cycle, look less complete. A small forward bias should be added to the test voltage range to remove the influence of possible incomplete switching for better confirming the change of remnant polarization during switching cycling.

2. There are confusing points on explanation about "the polarization evolution appears completely disconnected from the d_{33} evolution".

i) the authors claim that the samples are "almost wake-up free" and that "T-phase is not majority in HZO". And they attribute the d_{33} change during cycling to the transition from "a weakly-developed polar orthorhombic phase" to "a well-developed polar o-III phase with negative d_{33} ". The argument for this comes from the DFT calculation provided by the author. However, the DFT results in Fig. 4 is based on the tetra to ortho-III phase transition. This is not clearly consistent. Besides, when the curve in Fig. 4(c), which the authors claim "roughly goes as the d_{33} piezoelectric coefficient", is downward, the corresponding P_3 goes significant upward, which is different from the "almost wake-up free" phenomenon as claimed.

iii) More importantly, the authors mention that "in essence, $d_{33} \sim \epsilon_0 \chi_{33} \Delta \eta_3 / \Delta P_3$ ". Apparently, this correlates the polarization (corresponding to P_3) and d_{33} , which is different from "the polarization evolution appears completely disconnected from the d_{33} evolution". The authors should give more explanation about this.

3. We do not think the investigation of thicker HZO and others is not relevant here. The interface may play an important role for the phenomenon, if HZO thickness is changed, how about d_{33} ? In addition, investigation of amorphous HZO without annealing is another critical reference for confirming the results, because the process temperature is relatively low and we are not sure if amorphous part exist in the film and has an effect on the observed phenomenon.

Reviewer #2 (Remarks to the Author):

The authors replied to all my concerns. The results are interesting. The methods and procedures are clear and well-described. The conclusions are supported by the data, despite, obviously, other scenarios might explain them.

Reviewer #3 (Remarks to the Author):

The authors' response is generally reasonable to me. I understand that an exact understanding for the change of sign of d_{33} is still not easy yet. So I recommend publication of this paper.

Reviewer #1

The authors have addressed some of the issues pointed out in last time of review. However, some issues concerning the content and response are still unclear as below.

Comment 1. In Fig. 1(a) the authors give the P-V hysteresis loops of the W/HZO/W capacitor in the pristine state and after various numbers of ac field cycles. This is a very important information for the authors' discussion in response letter. However, the authors still need to provide some additional information to make things clearer:

i) there are some abnormal zig-zag curves in Fig. 1(a). The authors should give the corresponding I-V and V-t curves to help understanding the cause of abnormal jitter.

Response: This question was not in the first review. We would like to note that in the *P-V* measurements, there is always a measurement noise of the *measured* voltage compared to the *set* voltage (Fig. R1), which is a linear ramp of the triangular waveform. The jitter in the *P-V* loops in Fig. 1(a) of the manuscript is due to this measurement noise and does not have any physical meaning beyond that. A conventional approach of plotting the polarization vs the set voltage removes the jitter (see New Fig. 1(a) below). To address this reviewer request, we are also showing in the Supporting Information the conventionally looking I-V curves that were used to obtain the P-V curves (new Fig. S1).

Fig. R1. *I-t* and *V-t* data in the *P-V* measurements. There is a measurement noise of the measured voltage in comparison to the set voltage.

Note that while checking the P-V data, it was noticed that the area used for the calculation of the polarization was underestimated ($80 \times 80 \mu\text{m}^2$ while the nominal value was $95 \times 95 \mu\text{m}^2$). This was due to an incorrect calibration of the optical microscope used. The new

values measured are $93.5 \times 95.0 \mu\text{m}^2$, which are close to the nominal value. Hence, the polarization values indicated in the new Fig. 1(a) have been corrected and are slightly lower than those originally reported. This has, of course, no impact on the results and conclusions.

Since Reviewer 1 had not noticed that we had other measurements shown in the Supplementary Information, we want to make it clear in the manuscript text that several samples/devices were measured. Hence, rather than indicating one value for the pristine state and for the cycled ones (at 10^4 or 10^5 cycles), we have indicated a range of values that represent the different samples and devices measured for the d_{33} inversion study.

Modifications to the manuscript:

Figure 1: We have now the following Fig. 1(a) and included the I-V curves in the Supplementary Information.

New Fig. 1(a). Polarization vs set voltage.

Fig S1: I-V curves corresponding to the P-V loops shown in Fig. 1(a).

Page 4: “(the corresponding I-V curves are shown in Fig. S1 of the Supplementary Information). Additional examples of P-V loops upon cycling are given in the Supplementary Information.”

“Considering all samples/capacitors studied, P_r is typically of 19 - 23 $\mu\text{C}/\text{cm}^2$ in the pristine state, of 22 - 28 $\mu\text{C}/\text{cm}^2$ after 10^4 cycles and of $\sim 30 \mu\text{C}/\text{cm}^2$ after 10^5 cycles.”

Page 10: “From Figs. 4a and 4b, we find that the calculated maximum of η_3 for HZO corresponds to a polarization of about 33 $\mu\text{C}/\text{cm}^2$, which nearly falls within the range of polarization values measured in our capacitors in their pristine state (typically between 19 and 23 $\mu\text{C}/\text{cm}^2$), and after 10^5 cycles (30 $\mu\text{C}/\text{cm}^2$).”

Page 12 (Caption of Figure 4): “Interestingly, this range roughly overlaps with the one defined by the smallest ($\sim 19 - 23 \mu\text{C}/\text{cm}^2$ in the pristine state) and largest (30 $\mu\text{C}/\text{cm}^2$ after 10^5 cycles) remnant polarizations experimentally measured in our samples.”

ii) It can be easily found that the sample has around 0.5 V in-built electric field which makes the coercive field is no longer symmetric about 0. This also makes the forward switching, especially on the first cycle, look less complete. A small forward bias should be added to the test voltage range to remove the influence of possible incomplete switching for better confirming the change of remnant polarization during switching cycling.

Response: The question/request was not in the first review. We would like to mention that the horizontal shift of the P-V loops is not necessarily due to the built-in field, but it could be related to the imprint that can easily develop in the HZO capacitors even during the measurement cycle. It does not affect the completeness of switching for both polarities of the applied voltage, since the loops look saturated. However, even the presence of such field should not cast any doubt on the reported inversion of the d_{33} , which is observed not only by using the PFM imaging mode (Fig. 2) but also by the PFM spectroscopy (Fig. 3). We would like to emphasize that this effect is the main point of the manuscript while minor issues, such as P-V loop jitters and asymmetry, have no role in establishing this observation as a scientific fact.

Comment 2. There are confusing points on explanation about "the polarization evolution appears completely disconnected from the d_{33} evolution".

i) the authors claim that the samples are “almost wake-up free” and that “T-phase is not majority in HZO”. And they attribute the d_{33} change during cycling to the transition from “a weakly-developed polar orthorhombic phase” to “a well-developed polar o-III phase with negative d_{33} ”. The argument for this comes from the DFT calculation provided by the author. However, the DFT results in Fig. 4 is based on the tetra to ortho-III phase transition. This is not clearly consistent. Besides, when the curve in Fig. 4(c), which the authors claim “roughly goes as the d_{33} piezoelectric

coefficient”, is downward, the corresponding P_3 goes significant upward, which is different from the “almost wake-up free” phenomenon as claimed.

Response: We do not understand which points the Reviewer wants to make here and they were not in the first review.

By “almost wake-up free”, we means that, contrary to typical TiN/HZO/TiN capacitors, many of our W/HZO/W capacitors (and those shown in the manuscript) do not show anti-ferroelectric-like behavior in pristine or just after one cycle (by wake-up, it is usually meant that antiferroelectric loops need to be converted to ferroelectric one and this requires typically at least hundreds of cycles). Our HZO capacitors show already closed saturated ferroelectric P-V loops in the first P-V measurement. We then clearly state that the polarization does monotonically increase upon cycling (by about 20% upon 10^4 cycles – 10^5 cycles). Hence, our experimental results for P_3 (monotonically increase of P_r) are fully consistent with the behavior of Fig. 4(a) and our experimental results for the d_{33} evolution are fully consistent with the behavior of Fig. 4(c).

The DFT results in Fig. 4 cover the evolution from the paraelectric tetragonal phase (extreme left point noted “tetra”) to the (fully developed) ortho-III phase at the extreme right (noted ortho-III). All intermediate points represent a progressively developing orthorhombic polar phase (with a non-zero polarization) and not a tetragonal phase. The Reviewer seems to have missed this point. We had sketched by a dashed arrow in Fig. 4(b) and 4(c) the approximate region where we expect d_{33} to change sign in our HZO capacitors.

iii) More importantly, the authors mention that “in essence, $d_{33} \sim \epsilon_0 \chi_{33} \Delta \eta_3 / \Delta P_3$ ”. Apparently, this correlates the polarization (corresponding to P_3) and d_{33} , which is different from “the polarization evolution appears completely disconnected from the d_{33} evolution”. The authors should give more explanation about this.

We added: “the polarization evolution appears completely disconnected from the d_{33} evolution” at the request of Reviewer#2 in the revised manuscript. The expression $d_{33} \sim \epsilon_0 \chi_{33} \Delta \eta_3 / \Delta P_3$ indeed reflects that d_{33} does depend on the correlation between polarization ΔP_3 and strain $\Delta \eta_3$ changes. However, the piezoresponse does not grow with P_3 contrary to what happens in perovskite ferroelectrics. This is the point we wanted to emphasize with the sentence “the polarization evolution appears completely disconnected from the d_{33} evolution”. To avoid confusing or misleading the reader, we have changed the text as follow:

Modifications to the manuscript:

Abstract: $\text{Hf}_{0.5}\text{Zr}_{0.5}\text{O}_2$ capacitors subjected to ac cycling undergo a continuous transition from a positive effective piezoelectric coefficient d_{33} in the pristine state to a fully inverted negative d_{33} state **while, in parallel, the polarization monotonically increases.**

Page 4 (Introduction): We have removed the following sentence that we added (at the request of reviewer 2) in the first revised version: “They account for the surprising experimental observation that the sign of the polarization evolution upon ac cycling appears decoupled from the piezoelectric response evolution.”

To account for the monotonically increasing polarization (positive sign of the polarization evolution), we have added “while, in parallel, the polarization monotonically increases by about 20%”.

Page 6: We have removed the following sentence that we added (at the request of reviewer 2) in the first revised version: “indicating that the polarization evolution is decoupled from the piezoresponse evolution.”

Page 13 (Discussion): We modified the sentence to “Within this model, the non-monotonic development of η_3 with P_3 leads to the unusual behavior of d_{33} , whereby the piezoresponse does not grow with P_3 contrary to what happens in perovskite ferroelectrics.”

3. We do not think the investigation of thicker HZO and others is not relevant here. The interface may play an important role for the phenomenon, if HZO thickness is changed, how about d_{33} ? In addition, investigation of amorphous HZO without annealing is another critical reference for confirming the results, because the process temperature is relatively low and we are not sure if amorphous part exist in the film and has an effect on the observed phenomenon.

Response: We do think that the effect of thickness is not relevant for the paper. Indeed, it is very well known that the polarization in HZO is strongly dependent on the thickness. However, the response in d_{33} with thickness will not give information on the role of the interfaces in the phenomenon because reducing the thickness has also a strong impact on e.g. the strain imparted on the film upon the crystallization annealing (and hence on the stabilization of the polar phase itself, independently of the interfaces).

Here, we observed a new, never-before reported effect of electrically tunable piezoelectricity – this is the main point of the manuscript. It is a matter of separate studies to investigate the effect of thickness, composition, processing methods and other factors on manifestation of this phenomenon.

Regarding the question on the possible presence of an amorphous phase: **this is again a new question that was not asked in the first review.**

The P-V loops prove that there is negligible amount of the amorphous HZO, if any. It does not make sense that a negligible amount of the amorphous HZO significantly contributes to the electro-mechanical property (and not to the electrical one) and this would not explain the d_{33} inversion with electrical cycling.

Reviewer #2 (Remarks to the Author):

The authors replied to all my concerns. The results are interesting. The methods and procedures are clear and well-described. The conclusions are supported by the data, despite, obviously, other scenarios might explain them.

Reviewer #3 (Remarks to the Author):

The authors' response is generally reasonable to me. I understand that an exact understanding for the change of sign of d_{33} is still not easy yet. So I recommend publication of this paper.

REVIEWERS' COMMENTS

Reviewer #1 (Remarks to the Author):

The authors basically response most of my questions. Although I still have a concern on the final question, it can be studied in the next step.